# Expression of Adenosine Receptors in Rodent Pancreas

**DOI:** 10.3390/ijms20215329

**Published:** 2019-10-25

**Authors:** Mikio Hayashi

**Affiliations:** Department of Cell Physiology, Institute of Biomedical Science, Kansai Medical University, 2-5-1 shin-machi, Hirakata, Osaka 573-1010, Japan; hayashmi@hirakata.kmu.ac.jp; Tel.: +81-72-804-2322

**Keywords:** adenosine receptor, duct, ezrin, pancreas, secretion

## Abstract

Adenosine regulates exocrine and endocrine secretions in the pancreas. Adenosine is considered to play a role in acini-to-duct signaling in the exocrine pancreas. To identify the molecular basis of functional adenosine receptors in the exocrine pancreas, immunohistochemical analysis was performed in the rat, mouse, and guinea pig pancreas, and the secretory rate and concentration of HCO_3_^−^ in pancreatic juice from the rat pancreas were measured. The A_2A_ adenosine receptor colocalized with ezrin, an A-kinase anchoring protein, in the luminal membrane of duct cells in the mouse and guinea pig pancreas. However, a strong signal ascribed to A_2B_ adenosine receptors was detected in insulin-positive β cells in islets of Langerhans. The A_2A_ adenosine receptor agonist 4-[2-[[6-Amino-9-(*N*-ethyl-β-D-ribofuranuronamidosyl)-9*H*-purin-2-yl]amino]ethyl]benzenepropanoic acid (CGS 21680) stimulated HCO_3_^−^-rich fluid secretion from the rat pancreas. These results indicate that A_2A_ adenosine receptors may be, at least in part, involved in the exocrine secretion of pancreatic duct cells via acini-to-duct signaling. The adenosine receptors may be a potential therapeutic target for cancer as well as exocrine dysfunctions of the pancreas.

## 1. Introduction

The pancreas performs both exocrine and endocrine functions. Pancreatic acini secrete digestive enzyme- and Cl^−^-rich neutral fluid, and ducts secrete an HCO_3_^−^-rich pancreatic juice that neutralizes acidic chyme in the duodenum. The generally accepted model for HCO_3_^−^ transport involves electrogenic Cl^−^–HCO_3_^−^ exchangers (SLC26A) that operate in parallel with cAMP-activated Cl^−^ channels [cystic fibrosis transmembrane conductance regulator (CFTR)] and Ca^2+^-activated Cl^−^ channels (ANO1/TMEM16A) on the luminal membranes of duct cells [1,2,3,4]. Islets of Langerhans are involved in maintaining glucose homeostasis and comprise α-, β-, γ-, δ-, and ε-cells, which secrete glucagon, insulin and amylin, pancreatic polypeptide, somatostatin, and ghrelin, respectively [5].

Both exocrine and endocrine cells are regulated by nucleotides, as well as parasympathetic and sympathetic nerves and hormones [6]. Previous studies by Novak and coworkers demonstrated that the purinergic system has a coordinating function in acini-to-duct signaling [6]. Cholecystokinin stimulates the release of ATP, ectonucleoside triphosphate diphosphohydrolase (CD39), and ecto-5′-nucleotidase (CD73) from acini into pancreatic juice [7]. The ectonucleosides hydrolyze ATP to adenosine in the ductal lumen. Adenosine activates Cl^−^ conductance in pancreatic duct cells of the rat and guinea pig [8,9]. A pharmacological study indicated that A_2A_ adenosine receptors were involved in the water and HCO_3_^−^ secretory response in the dog pancreas [10]. In endocrine cells, N^6^-L-phenyl-isopropyl-adenosine, an A_1_ adenosine receptor agonist, reduced glucose-induced insulin secretion from the perfused rat and mouse pancreas [11,12,13]. In contrast, adenosine at 100 μM augmented insulin secretion through the A_2A_ adenosine receptor on isolated mouse islets [14].

A_1_, A_2A_, A_2B_, and A_3_ adenosine receptors are expressed in the rat and mouse pancreas [8,14,15]. Real-time PCR revealed that the rank order of the adenosine mRNA level was A_2A_ > A_2B_ ≥ A_3_ >> A_1_, where the level of A_2A_ was at least five times higher than of other receptors in the rat pancreas [8]. Immunohistochemical studies showed that A_2A_ and A_2B_ adenosine receptors were localized in the luminal membranes of rat duct cells and human duct cell lines [8,9]. Additionally, A_2A_ adenosine receptors were detected in the rat islets, most likely in β cells [8]. A_1_ and A_2A_ adenosine receptors were also found in isolated α-cells from mouse islets, and 4-[2-[[6-Amino-9-(*N*-ethyl-β-D-ribofuranuronamidosyl)-9*H*-purin-2-yl]amino]ethyl]benzenepropanoic acid (CGS 21680), an A_2A_ adenosine receptor agonist, stimulated glucagon secretion [16]. Protein expression of A_1_ adenosine receptors was identified in α-cells from the human pancreas [17]. However, the expression and function of adenosine receptors in primary human pancreatic duct cells are unknown.

A_2A_ and A_2B_ adenosine receptors generally increase cAMP levels, whereas A_1_ and A_3_ receptors decrease them [18]. Thus, the present study focused on functional A_2A_ and A_2B_ adenosine receptors involved in the ductal secretion of the pancreas. The results reveal that mouse and guinea pig duct cells express A_2A_ adenosine receptors. However, β cells in the islets express both A_2A_ and A_2B_ adenosine receptors. Furthermore, it was demonstrated that A_2A_ adenosine receptors contribute to exocrine secretion in the rat pancreas.

## 2. Results

### 2.1. A_2A_ and A_2B_ Receptors Expressed in Pancreatic Islets of the Rat

A previous study showed that A_2A_ adenosine receptors were expressed in the islets of the rat pancreas [8]. In order to determine the cells that expressed A_2A_ and A_2B_ adenosine receptors, triple staining of adenosine receptors was performed with insulin and glucagon on paraffin sections of the rat pancreas. The immunofluorescence of A_2A_ adenosine receptors was detected in insulin-positive β cells [19] (Figure 1A–C), but in few glucagon-positive α-cells (Figure 1D). Additionally, a strong signal ascribed to A_2B_ adenosine receptors was detected specifically in insulin-positive β cells (Figure 1E–H). Immunofluorescence was diminished with normal rabbit IgG for the isotype control (Figure 1I–L).

### 2.2. A_2A_ and A_2B_ Receptors Expressed in Pancreatic Islets of the Mouse

The immunolocalization of the A_2A_ and A_2B_ adenosine receptors was examined with paraffin sections of the mouse pancreas. Immunofluorescence ascribed to the A_2A_ adenosine receptors was localized on the inside of cells (Figure 2A). The A_2A_ adenosine receptors were expressed in insulin-positive β cells and glucagon-positive α-cells (Figure 2B–D). However, a strong signal ascribed to A_2B_ adenosine receptors was detected specifically in insulin-positive β cells (Figure 2E–H). The results were consistent with the distribution of A_2B_ adenosine receptors in the rat pancreas. Immunofluorescence was diminished with normal rabbit IgG for the isotype control (Figure 2I–L).

### 2.3. A_2A_ Receptors Expressed in the Pancreatic Ducts of Mice

Our previous study showed that both A_2A_ and A_2B_ adenosine receptors localized in the luminal membranes of rat duct cells [9]. In order to distinguish between pancreatic ducts and blood vessels, triple staining of adenosine receptors was performed with ezrin and PECAM-1 (platelet endothelial cell adhesion molecule-1) on paraffin sections of the mouse pancreas, as reported previously in the rat and guinea pig [4,9]. Immunofluorescence ascribed to the A_2A_ adenosine receptor was colocalized with ezrin, an A-kinase anchoring protein, to the luminal membrane of the pancreatic duct (Figure 3A–C). However, immunofluorescence ascribed to the A_2B_ adenosine receptor was weak in mouse ducts (Figure 3E–G). Additionally, the signal for A_2A_ and A_2B_ adenosine receptors was detected in PECAM-1-positive endothelial cells of blood vessels (Figure 3D,H). Immunofluorescence was diminished with normal rabbit IgG for the isotype control (Figure 3I–L).

### 2.4. A_2A_ Receptors Expressed in the Pancreatic Ducts of Guinea Pigs

We performed immunostaining of A_2A_ and A_2B_ adenosine receptors with ezrin and PECAM-1 on paraffin sections of the guinea pig. Immunofluorescence ascribed to the A_2A_ adenosine receptor was colocalized with ezrin to the luminal membrane of the pancreatic duct (Figure 4A–C). However, immunofluorescence ascribed to the A_2B_ adenosine receptor was weak in guinea pig ducts (Figure 4 E–G). The signal for A_2B_ adenosine receptors was detected in the PECAM-1-positive endothelial cells of blood vessels (Figure 4H). Immunofluorescence was diminished with normal rabbit IgG for the isotype control (Figure 4I–L).

### 2.5. Expression of A_2A_ and A_2B_ Receptor Protein in Rat Pancreatic Ducts

We next performed Western blot analysis to examine the expression of A_2A_ and A_2B_ adenosine receptors in the rat pancreatic ducts. We detected A_2A_ (ADORA2A, 54 kDa) and A_2B_ (ADORA2B, 55 kDa) adenosine receptors in the lysates of the isolated ducts (Figure 5; *n* = 2 rats). In addition, the A_2A_ and A_2B_ adenosine receptors were detected in the lysates of Capan-1, which is a human pancreas adenocarcinoma cell line, but not in HEK293 cells (*n* = 2).

### 2.6. A_2A_ Receptor Agonist Elicited Pancreatic Secretion in Rats

In order to demonstrate whether adenosine regulated exocrine secretion, the secretory rate and concentration of protein and HCO_3_^−^ in pancreatic juice from the rat pancreas were measured. Specific adenosine receptor agonists were tested to identify functional adenosine receptors. The intravenous injection of CGS 21680 (20 nmol/kg body weight), an A_2A_ adenosine receptor agonist, significantly increased the secretory rate from 0.40 ± 0.05 in the control to 0.72 ± 0.09 μL/min after 20 min and sustained it for 20 min (Figure 6A; *n* = 6 rats). The concentration of protein in pancreatic juice was decreased from 77.7 ± 8.4 to 41.2 ± 5.5 g/L after 40 min, indicating ductal secretion (Figure 6B). In addition, the HCO_3_^−^ concentration was increased from 38.2 ± 3.1 to 52.7 ± 5.6 mM after 40 min, indicating HCO_3_^−^-rich ductal secretion (Figure 6C). In contrast, 2-({6-Amino-3,5-dicyano-4-[4-(cyclopropylmethoxy)phenyl]pyridin-2-yl}sulfanyl)acetamide (BAY 60-6583, 20 nmol/kg body weight), an A_2B_ adenosine receptor agonist, had a negligible effect on the secretory rate: 0.59 ± 0.08 μL/min in the control and 0.63 ± 0.06 μL/min with BAY 60-6583 (Figure 6D; *P* = 0.72, *n* = 5 rats). However, the protein concentration showed a tendency to decrease from 102.9 ± 14.8 to 65.5 ± 5.9 g/L (Figure 6E; *P* = 0.07), indicating ductal secretion. The HCO_3_^−^ concentration was slightly increased from 31.3 ± 3.4 to 38.3 ± 2.4 mM (Figure 6F; *P* = 0.30). In the control experiment, secretin (Sec, 0.1 nmol/kg body weight) significantly increased the secretory rate and HCO_3_^−^ concentration in pancreatic juice, indicating that ducts secreted an HCO_3_^−^-rich fluid, as reported previously [20] (Figure 6A,C). In addition, cholecystokinin (CCK, 0.3 nmol/kg body weight) increased the secretory rate and protein concentration, but decreased the HCO_3_^−^ concentration, indicating that acini secreted digestive enzyme- and Cl^−^-rich neutral fluid [21] (Figure 6A–C). The vehicle control (0.4% DMSO in saline) did not influence exocrine secretion for 40 min (Figure 6G–I; *n* = 3 rats).

### 2.7. Effect of Adenosine Receptor Antagonists on Pancreatic Secretion in Rats

Cholecystokinin stimulates the release of ATP and ectonucleosides from acini into pancreatic juice [7]. Adenosine is produced by the hydrolysis of ATP in the ductal lumen. In order to demonstrate whether luminal adenosine regulated adenosine receptors, specific antagonists were used. The moderate concentration of cholecystokinin (CCK, 0.1 nmol/kg body weight) increased the secretory rate and protein concentration, as reported previously [21] (Figure 7A,B; *n* = 4 rats). The response to CCK was reproducible based on repeated applications in the vehicle control experiments. In preliminary experiments, the intravenous injection of 2-(2-Furanyl)-7-[3-(4-methoxyphenyl)propyl]-7*H*-pyrazolo[4,3-*e*][1,2,4]triazolo[1,5-*c*]pyrimidin-5-amine (SCH-442416; 10 nmol/kg body weight), an A_2A_ adenosine receptor antagonist, slightly decreased the secretory rate to 74.2 ± 0.2% of the first stimulation (Figure 7D; *P* = 0.36, *n* = 2 rats). Additionally, the intravenous injection of 8-[4-[4-(4-chlorophenzyl)piperazide-1-sulfonyl)phenyl]]-1-propylxanthine (PSB 603; 10 nmol/kg body weight), an A_2B_ adenosine receptor antagonist, slightly decreased the secretory rate to 76.4 ± 7.1% (Figure 7G; *P* = 0.31, *n* = 5 rats). Neither SCH-442416 nor PSB 603 led to an increase in the concentration of protein (Figure 7E,H) or had a significant effect on the HCO_3_^−^ concentration (Figure 7C,F,I).

### 2.8. Expression of ADORA2B Associated with Poor Prognosis of Pancreatic Adenocarcinoma

In contrast to the rat pancreas, the A_2B_ adenosine receptors showed the highest mRNA levels among four subtypes in human pancreatic adenocarcinoma cell lines [8]. To determine whether the expression of adenosine receptors was associated with a poor prognosis of pancreatic adenocarcinoma patients, we conducted statistical analysis using The Cancer Genome Atlas (TCGA) database. Analysis of RNA-sequencing data of pancreatic adenocarcinoma (TCGA, Provisional) revealed that a low expression of *ADORA2A*, which encodes the A_2A_ adenosine receptor, was associated with poor overall survival and disease-free survival (Figure 8A,B; log-rank *P* = 0.0302 and *P* = 0.0104, respectively). In contrast, high expression of *ADORA2B* (A_2B_ adenosine receptor) was associated with poor disease-free survival (Figure 8D; log-rank *P* = 0.0125). However, the expression of *ADORA1* and *ADORA3* was not associated with the prognosis of pancreatic adenocarcinoma patients (log-rank *P* = 0.310 and *P* = 0.322, respectively).

## 3. Discussion

In the present study, it was demonstrated that the A_2A_ adenosine receptor contributed to exocrine secretion in the rodent pancreas. This conclusion was based on the following major results: the A_2A_ adenosine receptor agonist stimulated a HCO_3_^−^-rich fluid secretion from the rat pancreas (Figure 6A–C); the A_2A_ adenosine receptor colocalized with ezrin in the luminal membrane of duct cells in the mouse and guinea pig pancreas (Figure 3 and Figure 4), as reported in rats previously. The conclusion is consistent with a pharmacological study in the dog pancreas [10].

### 3.1. Compartmentalization of cAMP Signaling in the Luminal Regions of Pancreatic Duct Cells

In pancreatic ducts, adenosine is produced by the hydrolysis of ATP that is secreted from acini by CCK stimulation [6]. Adenosine binds to adenosine receptors on the luminal membrane of duct cells and stimulates ductal secretion [22]. In the present study, A_2A_ and A_2B_ adenosine receptor antagonists showed a tendency to suppress 25% of the secretory rate by CCK stimulation (Figure 7), indicating acini-to-duct signaling. A_2A_ and A_2B_ adenosine receptors primarily signal via G_s_ proteins, resulting in the activation of adenylyl cyclase, an increase in cAMP production, activation of a membrane-associated isoform of protein kinase A (type II PKA), and subsequent activation of cAMP-activated Cl^−^ channels (CFTR) [23,24,25]. Previous studies reported that ezrin forms the scaffold for type II PKA and cAMP signaling compartments, including the A_2B_ adenosine receptor, adenylyl cyclase, and CFTR [26,27,28]. The membrane-associated adenylyl cyclase isoforms (AC3, AC4, AC6, AC7, and AC9) were found to be expressed in the mouse pancreas [29]. Furthermore, AC6 was shown to play a critical role in cAMP/PKA-mediated signaling in pancreatic exocrine cells. The AC6 physically and functionally associates with CFTR at the apical surface of intestinal epithelial cells [30]. A recent study reported that the stimulation of A_2A_ adenosine receptors activates AC6, which is bound to A-kinase-anchoring protein (AKAP79/150), to synthesize cAMP that is used by PKA and phosphodiesterase 3A (PDE3A) in hepatocytes [31]. Further studies are required to clarify whether A_2A_ adenosine receptors associate with AC6 and lead to the compartmentalization of cAMP signaling in the luminal regions of pancreatic duct cells.

### 3.2. Role of A_2A_ and A_2B_ Adenosine Receptors in Pancreas

In accordance with the present results, previous studies demonstrated that A_2A_ adenosine receptors regulated anion secretion in the gerbil middle ear epithelium [32], rabbit distal bright convoluted tubule cell line [33], and mouse colonic epithelia [34]. In addition to epithelial transport, the A_2A_ adenosine receptor is known to be involved in the endocrine pancreas [13]. In the present study, A_2A_ and A_2B_ adenosine receptors were detected in insulin-positive β cells (Figure 1 and Figure 2). A previous study demonstrated that A_2A_ adenosine receptors stimulated insulin secretion on mouse islets [14]. However, the A_2B_ adenosine receptor antagonists were shown to increase plasma insulin levels in rats [35]. Future studies are required to clarify the intracellular signaling in pancreatic β cells.

### 3.3. Composition of Adenosine Receptor Subtype in Duct Cells

Several limitations of this study should be acknowledged. The protein level of A_2A_ and A_2B_ adenosine receptors could not be quantified by immunohistochemical analysis. The effects of CGS 21680 and BAY 60-6583 were small compared to those of secretin in in vivo experiments (Figure 6). Only the maximum dose of the chemical was tested. In the present study, ductal secretion was stimulated significantly more by CGS 21680 than BAY 60-6583 at the same concentration, indicating that A_2A_ adenosine receptors were dominant in rat duct cells, whereas BAY 60-6583 showed a tendency to promote HCO_3_^−^-rich fluid secretion (Figure 6E,F). Thus, we cannot rule out the existence of A_2B_ adenosine receptors in pancreatic duct cells. Previous studies postulated that A_2A_ and A_2B_ adenosine receptors form functional heterooligomers [36]. The affinities of the A_2A_ adenosine receptors for adenosine or CGS 21680 were shown to be decreased on co-expression with A_2B_ adenosine receptors in recombinant cells [37]. Meanwhile, the affinity of A_2B_ adenosine receptors for BAY 60-6583 was not altered by co-expression with A_2A_ adenosine receptors. Future studies are needed in order to establish the presence of the heterooligomer in pancreatic duct cells and its functional relevance.

### 3.4. A Potential Therapeutic Target for Pancreatic Cancer

In contrast to the present results, we demonstrated that adenosine regulated anion secretion via A_2B_ adenosine receptors in the luminal membrane of Capan-1, which is a human pancreas adenocarcinoma cell line [9]. The A_2B_ adenosine receptors had the highest mRNA level among four subtypes in human pancreatic adenocarcinoma cell lines (PANC-1 and CFPAC-1) [8]. In silico analysis of TCGA data showed that a high mRNA expression of *ADORA2B* was associated with a poor disease-free survival of pancreatic adenocarcinoma patients (Figure 8D). However, a low expression of *ADORA2A* was associated with poor prognosis (Figure 8A,B). Similarly, the expression of A_2B_ adenosine receptors was highest in prostate cancer cells and bladder urothelial carcinoma [38,39]. Moreover, mRNA and the protein expression of A_2B_ adenosine receptors were consistently upregulated in bladder urothelial and colorectal carcinoma compared with normal tissues [39,40]. The high expression of A_2B_ adenosine receptors was associated with a poor prognosis in bladder urothelial carcinoma and breast cancer patients [39,41]. The expression of A_2B_ adenosine receptors was increased in colon and breast cancer cells by hypoxia, suggesting a potential therapeutic target for cancer [40,42]. Indeed, selective A_2B_ adenosine receptor antagonists inhibited the proliferation of prostate, colon, and breast cancer cells [38,40,43]. The blockade of A_2B_ adenosine receptors was shown to increase the sensitivity of mouse GL261 glioma cells to the chemotherapeutic drug temozolomide [44]. In addition, the selective antagonist improved the survival of mice bearing metastatic breast tumors [41]. Future studies are needed in order to verify the presence of A_2B_ adenosine receptors in primary pancreatic tumors and confirm their pathophysiological functions related to proliferation, metastasis, and chemoresistance in pancreatic cancer.

## 4. Materials and Methods

### 4.1. Experimental Animals

Male Sprague–Dawley rats [specific pathogen free (SPF), 12–14 weeks old, 320–430 g], male Jcl:ICR mice (SPF, 8 weeks old, 36–38 g), and female Hartley guinea pigs (SPF, 4–8 weeks old, 300–420 g) were housed in the animal care facility of the Laboratory Animal Center, Kansai Medical University, Japan, under a 12-h light–dark cycle (lights on at 8:00). Constant temperature (21–23 °C) and relative humidity (40–60%) were maintained. The animals had free access to standard diet and water. The care of the animals and experimental procedures were approved by the Animal Experimentation Committee of Kansai Medical University (approval numbers 13-027 of 14 February 2013, 14-010 of 27 February 2014, and 15-106 of 25 May 2015). Every effort was made to reduce the number of the animals used and minimize animal suffering.

### 4.2. Preparation of Paraffin-Embedded Sections

The rats were anesthetized with isoflurane and a mixture of medetomidine [0.375 mg/kg body weight (b.w.) intraperitoneal (i.p.)], midazolam (2.0 mg/kg b.w.), and butorphanol (2.5 mg/kg b.w.). The mice were anesthetized with isoflurane and a mixture of medetomidine (0.3 mg/kg body weight i.p.), midazolam (4.0 mg/kg b.w.), and butorphanol (5.0 mg/kg b.w.). The guinea pigs were anesthetized with isoflurane and a mixture of medetomidine (0.5 mg/kg body weight i.p.), midazolam (5.0 mg/kg b.w.), and butorphanol (2.5 mg/kg b.w.). The anesthetized animals were perfused transcardially with 4% paraformaldehyde in phosphate-buffered saline (PBS). The pancreas was fixed with 4% paraformaldehyde in PBS for 24 h, embedded in paraffin, and sectioned.

### 4.3. Immunohistochemistry

Detailed methods of immunohistochemistry were previously described [9]. Briefly, autofluorescence was blocked by 0.1 M Tris-glycine. Non-specific binding was blocked with 2% normal serum in PBS. Preparations were subsequently incubated with primary antibodies for A_2A_ and A_2B_ adenosine receptors (ADORA2A and ADORA2B), glucagon, insulin, ezrin, and PECAM-1 (platelet endothelial cell adhesion molecule-1) in immunoreaction enhancer solution (Can Get Signal immunostain; Toyobo, Osaka, Japan) overnight at 4 °C (Table 1). Secondary antibodies conjugated to Alexa488, Alexa568, or Alexa647 (1:400; Molecular Probes) were added for 1 h. As the isotype control, normal rabbit IgG (2 μg/mL; Wako Pure Chemical, Osaka, Japan) was used, and scanning was performed using the same settings. Nuclei were stained with 4′,6-diamidino-2-phenylindole (DAPI) at 1 μg/mL. Fluorescence was observed with a confocal laser scanning microscope (LSM510 META; Carl Zeiss, Oberkochen, Germany).

### 4.4. Western Immunoblotting

Male Sprague–Dawley rats were sacrificed by cervical dislocation. Pancreatic ducts were isolated by enzymatic digestion and microdissection from the pancreas, as previously described [1]. The pancreas was removed and digested with collagenase (Type IV, 124 U/mL; Worthington Biochemical, Lakewood, NJ, USA) and trypsin inhibitor (0.01%; Sigma, St. Louis, MO, USA) in Tyrode solution at 37 °C for 1 h with vigorous shaking. Tyrode solution contained the following (in mM): 140 NaCl, 0.33 NaH_2_PO_4_, 5.4 KCl, 1.8 CaCl_2_, 0.5 MgCl_2_, 5 4-(2-hydroxyethyl)-1-piperazineethanesulfonic acid (HEPES), and 5.5 D-glucose; pH was adjusted to 7.4 with NaOH. Interlobular and intralobular ducts (outside diameter of 20–60 μm) were microdissected under a stereomicroscope (M205 C; Leica Microsystems, Wetzlar, Germany).

The ducts were washed with cold PBS, treated with trichloroacetic acid (10%) on ice for 30 min, and then centrifuged. The pellet was solubilized in lysis buffer containing urea (9 M), Triton X-100 (2%), dithiothreitol (1%), and lithium dodecyl sulfate (2%). The samples (30 μg/lane protein) were fractionated on SDS polyacrylamide gel (7.5%), electroblotted onto polyvinylidene fluoride (PVDF) membranes (Merck Millipore, Burlington, MA, USA), blocked with skim milk (1%), and reacted with anti-ADORA2A (1:200) or anti-ADORA2B (1:1000) antibodies (Table 1). The reaction was visualized with a secondary antibody labeled with alkaline phosphatase (Promega, Madison, WI, USA).

Capan-1 cells (#HTB-79; ATCC) were grown in Iscove’s modified Dulbecco’s medium with Glutamax (Gibco; Invitrogen, Carlsbad, CA, USA) and 20% fetal bovine serum (FBS) (ATCC). Cells were grown to confluent monolayers on membranes (Transwell, #3450; Corning Inc., Corning, NY, USA). HEK293 cells (Clontech Laboratories, Mountain View, CA, USA) were cultured in Dulbecco’s modified Eagle medium (Sigma). Protein samples were extracted from the cells as described above.

### 4.5. In Vivo Collection of Pancreatic Secretion

Male Sprague–Dawley rats were left overnight without food, but with access to water. The rats were anesthetized with isoflurane, pentobarbital (25 mg/kg body weight i.p.), and ethyl carbamate (1 g/kg b.w.) to maintain a stable depth of anesthesia without awareness episodes. The animals were placed on a heating table to maintain body temperature. The surgical technique used to collect pancreatic juice from the rat with a slight modification is described elsewhere [20,45]. The abdomen was opened by a midline incision. The common pancreatic bile duct was cannulated, and pancreatic juice was collected in a silicone tube (CP-N 0.5×1; Shin-Etsu Polymer, Tokyo, Japan). In order to collect pancreatic juice, free of bile, the proximal end of the bile duct was ligated and cut. For intravenous injections, a polyethylene catheter was placed in one of the external jugular veins. Sample volumes were determined by the length of pancreatic juice in the silicone tube. Samples were collected into 1 mL of saline, and the protein concentration was measured at OD280 with a spectrophotometer (BioPhotometer; Eppendorf, Hamburg, Germany). The samples were equilibrated with 5% CO_2_ overnight and pH values were measured with a pH meter (B-71X; Hobiba, Kyoto, Japan) at 37 °C in a CO_2_ incubator. The HCO_3_^−^ concentration was calculated with the Henderson–Hasselbalch equation and determined with the calibration curve method.

### 4.6. In Silico Analysis of TCGA Data

Statistical analysis was performed on data from The Cancer Genome Atlas (TCGA) database using the online cBioPortal for Cancer Genomics platform (http://www.cbioportal.org) [46,47]. A total of 179 cases were selected, corresponding to mRNA data (RNA Seq V2 RSEM) from Pancreatic adenocarcinoma patients (TCGA, Provisional). A cohort of pancreatic adenocarcinoma patients was divided into those with high (Z-score >1.5) and low expressions of mRNA. Kaplan–Meier analysis with the log-rank test was performed to estimate the overall survival and disease/progression-free survival.

### 4.7. Materials

CGS 21680, BAY 60-6583, and PSB 603 were obtained from Tocris Bioscience (Bristol, UK). SCH-442416 was obtained from Sigma. Rat secretin was obtained from GenScript (Piscataway, NJ, USA). Cholecystokinin (CCK8 sulfated form) was obtained from Peptide Institute (Ibaraki, Japan).

### 4.8. Statistics

Means ± SEM of the number *n* of experiments are shown. Significance of the differences was analyzed by the one-way analysis of variance, with *p* < 0.05 indicating significance. Data were analyzed in Igor or Microsoft Excel.

## 5. Conclusions

The results showed that A_2A_ adenosine receptors may be, at least in part, involved in the exocrine secretion of pancreatic duct cells via acini-to-duct signaling. The A_2A_ and A_2B_ adenosine receptors may be a potential therapeutic target for cancer as well as exocrine dysfunctions of the pancreas.

## Figures and Tables

**Figure 1 ijms-20-05329-f001:**
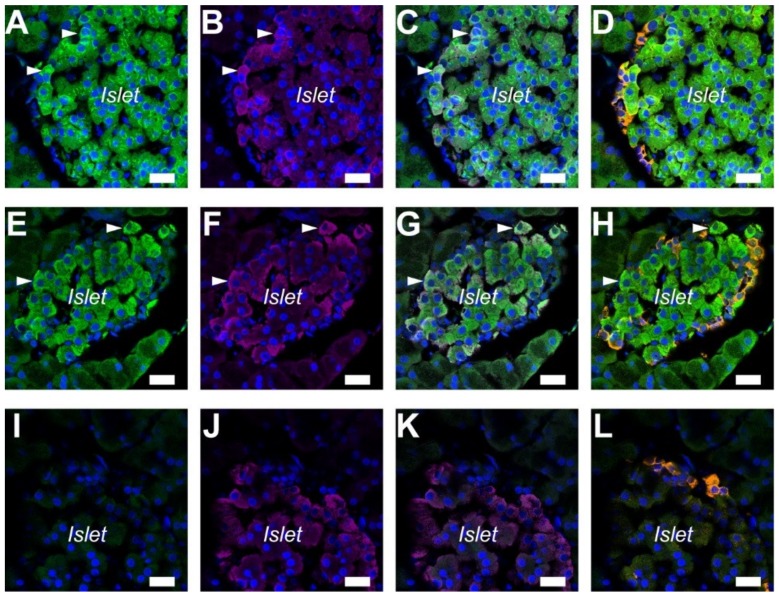
Immunolocalization of A_2A_ and A_2B_ adenosine receptors in the pancreatic islets of the rat. (**A**) Fluorescence of A_2A_ adenosine receptor in the cell membrane (arrowheads). (**B**) Fluorescence image of insulin. (**C**) Overlay image of **A** and **B**. (**D**) Overlay of **A** and fluorescence image of glucagon (orange) in the same sample. (**E**) Fluorescence of A_2B_ adenosine receptor in cells (arrowheads). (**F**) Fluorescence image of insulin. (**G**) Overlay image of **E** and **F**. (**H**) Overlay of **E** and fluorescence image of glucagon (orange). (**I**) Isotype control image of the rat pancreas with normal rabbit IgG (2 μg/mL). (**J**) Fluorescence image of insulin. (**K**) Overlay image of **I** and **J**. (**L**) Overlay of **I** and fluorescence image of glucagon (orange). 4′,6-diamidino-2-phenylindole (DAPI) was used to stain nuclei (blue). Representative images of three rats are shown. Bars = 20 μm.

**Figure 2 ijms-20-05329-f002:**
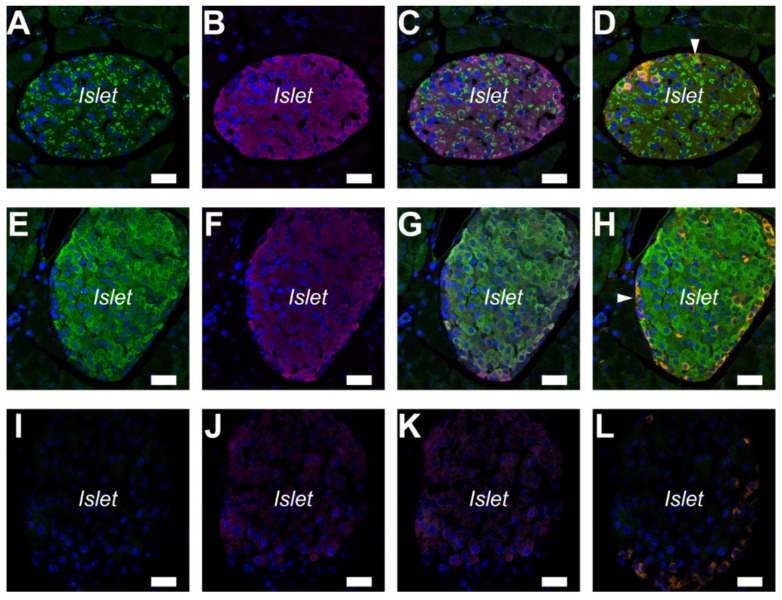
Immunolocalization of A_2A_ and A_2B_ adenosine receptors in pancreatic islets of the mouse. (**A**) Fluorescence of A_2A_ adenosine receptor inside cells. (**B**) Fluorescence image of insulin. (**C**) Overlay image of **A** and **B**. (**D**) Overlay of **A** and fluorescence image of glucagon (orange) in the same sample. A few glucagon-positive α-cells expressed A_2A_ adenosine receptors (arrowhead). (**E**) Fluorescence of A_2B_ adenosine receptor in cells. (**F**) Fluorescence image of insulin. (**G**) Overlay image of **E** and **F**. (**H**) Overlay of **E** and fluorescence image of glucagon (*orange*). Glucagon-positive α-cells did not express A_2B_ adenosine receptors (arrowhead). (**I**) Isotype control image of the mouse pancreas with normal rabbit IgG (2 μg/mL). (**J**) Fluorescence image of insulin. (**K**) Overlay image of **I** and **J**. (**L**) Overlay of **I** and fluorescence image of glucagon (orange). DAPI was used to stain nuclei (blue). Representative images of three mice are shown. Bars = 20 μm.

**Figure 3 ijms-20-05329-f003:**
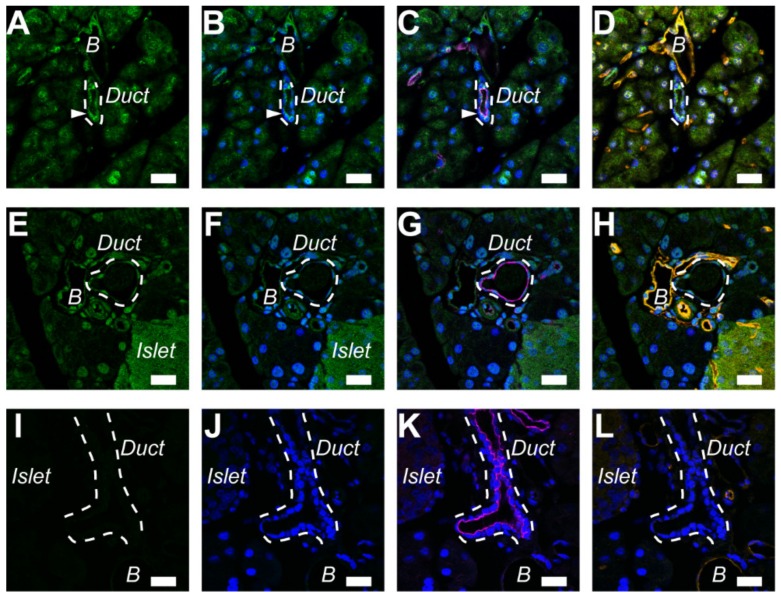
Immunolocalization of A_2A_ adenosine receptors in the pancreatic duct of the mouse. (**A**) Fluorescence of A_2A_ adenosine receptor in the luminal membrane of duct cells (arrowhead). The broken lines indicate a duct. The signal was detected in the endothelial cells of blood vessels (B). (**B**) Overlay of **A** and nuclear staining with DAPI (blue). (**C**) Overlay of **B** and fluorescence image of ezrin (purple). (**D**) Overlay of **B** and fluorescence image of a blood vessel marker (orange: platelet endothelial cell adhesion molecule-1, or PECAM-1) in the same sample. (**E**) The signal for the A_2B_ adenosine receptor was detected in the blood vessels (B) and islet. The broken line indicates a duct. (**F**) Overlay of **E** and nuclear staining. (**G**) Overlay of **F** and fluorescence image of ezrin (purple). (**H**) Overlay of **F** and fluorescence image of PECAM-1 (orange). (**I**) Isotype control image of the mouse pancreas with normal rabbit IgG (2 μg/mL). (**J**) Overlay of **I** and nuclear staining. (**K**) Overlay of **J** and fluorescence image of ezrin (purple). (**L**) Overlay of **J** and fluorescence image of PECAM-1 (orange). Representative images of three mice are shown. Bars = 20 μm.

**Figure 4 ijms-20-05329-f004:**
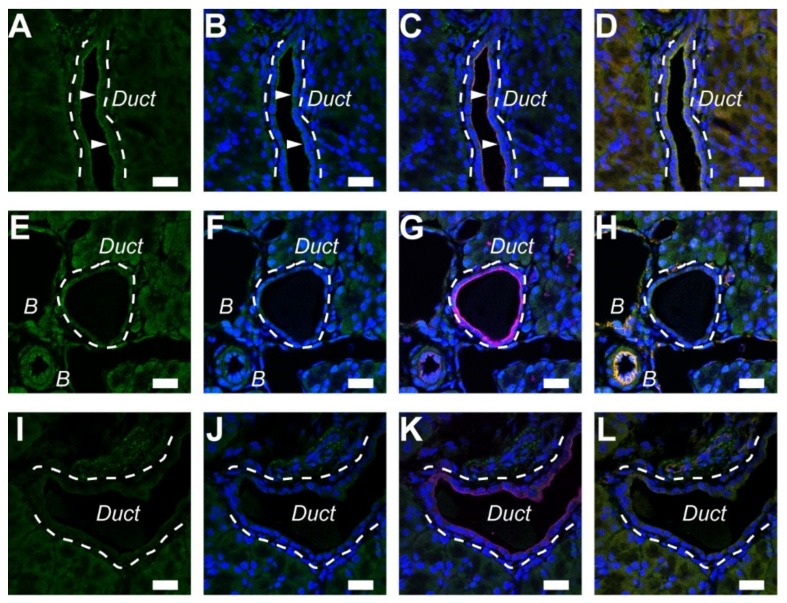
Immunolocalization of A_2A_ adenosine receptors in the pancreatic duct of the guinea pig. (**A**) Fluorescence of the A_2A_ adenosine receptor in the luminal membrane of duct cells (arrowheads). The broken lines indicate a duct. (**B**) Overlay of **A** and nuclear staining with DAPI (blue). (**C**) Overlay of **B** and fluorescence image of ezrin (purple). (**D**) Overlay of **B** and fluorescence image of a blood vessel marker (orange: PECAM-1) in the same sample. (**E**) The signal for the A_2B_ adenosine receptor was detected in the blood vessels (B). The broken line indicates a duct. (**F**) Overlay of **E** and nuclear staining. (**G**) Overlay of **F** and fluorescence image of ezrin (purple). (**H**) Overlay of **F** and fluorescence image of PECAM-1 (orange). (**I**) Isotype control image of the guinea pig pancreas with normal rabbit IgG (2 μg/mL). (**J**) Overlay of **I** and nuclear staining. (**K**) Overlay of **J** and fluorescence image of ezrin (purple). (**L**) Overlay of **J** and fluorescence image of PECAM-1 (orange). Representative images of three guinea pigs are shown. Bars = 20 μm.

**Figure 5 ijms-20-05329-f005:**
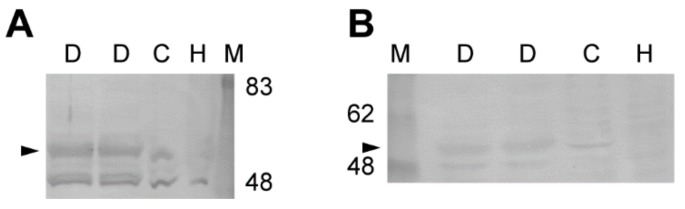
Immunoblot of A_2A_ (**A**) and A_2B_ (**B**) adenosine receptors from the rat pancreatic duct. Protein samples were resolved by SDS-PAGE. Arrowheads indicate adenosine receptor proteins detected by immunoblotting using anti-ADORA2A (**A**, 1:200, sc-13937) or anti-ADORA2B (**B**, 1:1000, AAR-003) antibody. Representative membranes from two independent experiments are shown. M, marker; D, duct; C, Capan-1; H, HEK293.

**Figure 6 ijms-20-05329-f006:**
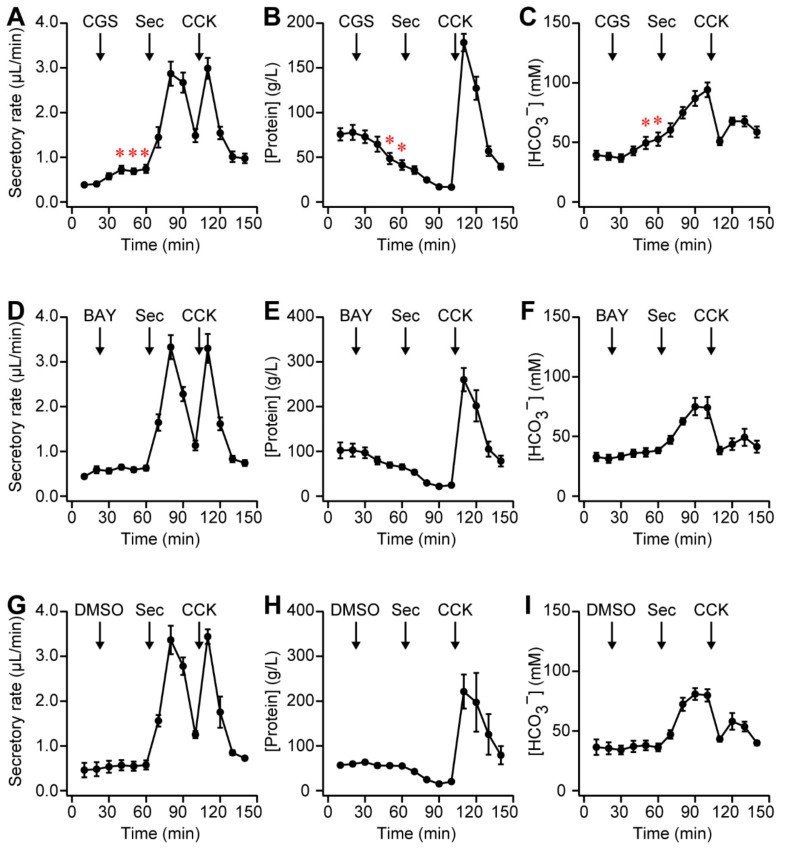
(**A**) Time-course of secretory rate of pancreatic juice from the anesthetized rats, which were intravenously injected with 4-[2-[[6-Amino-9-(*N*-ethyl-β-D-ribofuranuronamidosyl)-9*H*-purin-2-yl]amino]ethyl]benzenepropanoic acid (CGS 21680; CGS, 20 nmol/kg body weight), secretin (Sec, 0.1 nmol/kg body weight), and cholecystokinin (CCK, 0.3 nmol/kg body weight), as indicated by arrows (*n* = 6 rats, * *p* < 0.05). Values were compared with the control value at 20 min. Pancreatic juice was collected in a silicone tube. Sample volumes were determined by the length of pancreatic juice in the silicone tube. The concentrations of protein (**B**) and HCO_3_^−^ (**C**) in pancreatic juice. (**D**–**F**) Time-courses of experiments in the anesthetized rats, which were intravenously injected with 2-({6-Amino-3,5-dicyano-4-[4-(cyclopropylmethoxy)phenyl]pyridin-2-yl}sulfanyl)acetamide (BAY 60-6583; BAY, 20 nmol/kg body weight), secretin, and cholecystokinin (*n* = 5 rats). (**G**–**I**) Time-courses of experiments in the anesthetized rats, which were intravenously injected with vehicle control (0.4% DMSO in saline, 1 mL/kg body weight), secretin, and cholecystokinin (*n* = 3 rats). Secretin increased the secretory rate (**A**,D****) and HCO_3_^−^ concentration in pancreatic juice (**C**,**F**), indicating ductal secretion. Cholecystokinin increased the secretory rate and protein concentration (**B**,**E**), indicating acinar secretion.

**Figure 7 ijms-20-05329-f007:**
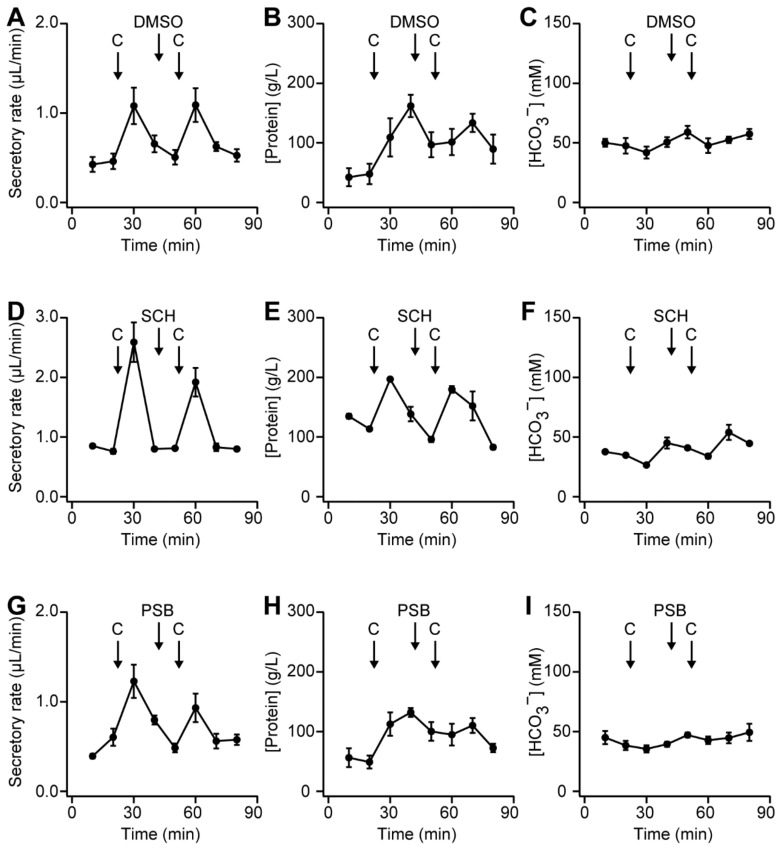
(**A**) Time-course of secretory rate of pancreatic juice from the anesthetized rats, which were intravenously injected with cholecystokinin (CCK, 0.1 nmol/kg body weight) and vehicle control (0.1% DMSO in saline, 1 mL/kg body weight), indicated by arrows (*n* = 4 rats). The concentrations of protein (**B**) and HCO_3_^−^ (**C**) in pancreatic juice. (**D**–**F**) Time-courses of experiments in the anesthetized rats, which were intravenously injected with cholecystokinin and 2-(2-Furanyl)-7-[3-(4-methoxyphenyl)propyl]-7*H*-pyrazolo[4,3-*e*][1,2,4]triazolo[1,5-*c*]pyrimidin-5-amine (SCH-442416; SCH, 10 nmol/kg body weight) (*n* = 2 rats). (**G**–**I**) Time-courses of experiments in the anesthetized rats, which were intravenously injected with cholecystokinin and 8-[4-[4-(4-chlorophenzyl)piperazide-1-sulfonyl)phenyl]]-1-propylxanthine (PSB 603; PSB, 10 nmol/kg body weight) (*n* = 5 rats). Neither SCH-442416 nor PSB 603 increased the concentrations of protein (**E**,**H**), and both had a negligible effect on the HCO_3_^−^ concentration (**F**,**I**).

**Figure 8 ijms-20-05329-f008:**
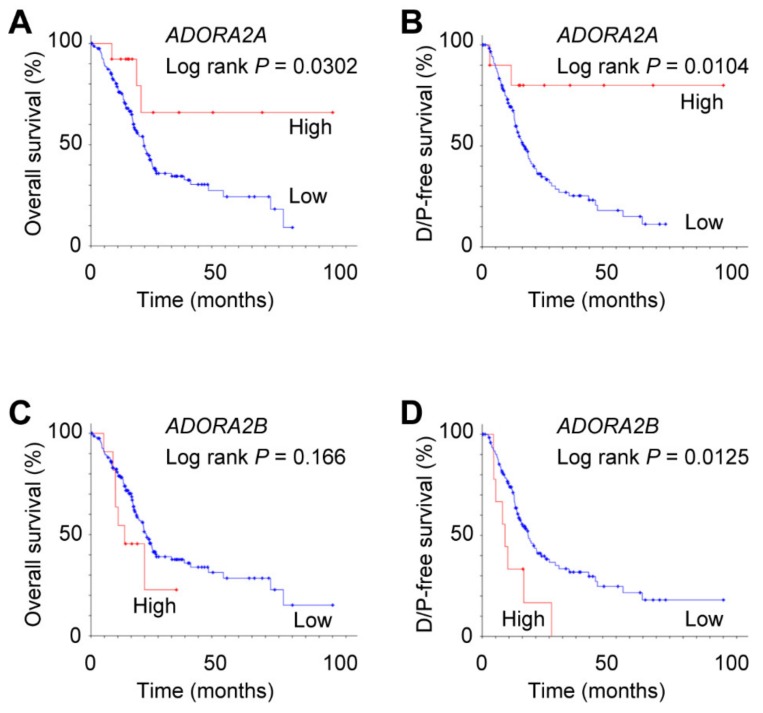
Association of mRNA expression of *ADORA2A* and *ADORA2B* with overall survival (**A**,**C**) and disease/progression (D/P)-free survival (**B,D**) in pancreatic adenocarcinoma patients. (**A**) High (*n* = 13) and low (*n* = 165) expression of *ADORA2A* (*P* = 0.0302). (**B**) High (*n* = 10) and low (*n* = 128) expression of *ADORA2A* (*P* = 0.0104). (**C**) Median survival time was 12.5 and 20.6 months with high (*n* = 12) and low (*n* = 166) expression of *ADORA2B*, respectively (*P* = 0.166). (**D**) Median disease-free survival time was 8.5 and 17.1 months with high (*n* = 10) and low (*n* = 128) expression of *ADORA2B*, respectively (*P* = 0.0125).

**Table 1 ijms-20-05329-t001:** Antibodies used in immunohistochemistry.

Protein Accession	Antigen	Correlation			Dilution	Catalogue Number Manufacturer
		Rat	Mouse	Guinea Pig		
ADORA2A NP_000666	331–412	57%	62%	78%	1:100	sc-13937 Santa Cruz
ADORA2B NP_000667	147–166	70%	65%	85%	1:800	AAR-003 Alomone Labs
Glucagon NP_999489	53–81	100%	100%	N/A	1:8000	G2654 Sigma-Aldrich
insulin NP_000198	recombinant	92%	92%	N/A	1:2000	Y370 Yanaihara Institute
ezrin NP_001104547	362–585	96%	96%	93%	1:200	MS-661 Lab Vision
PECAM-1 NP_001027550	C-terminus				1:400	sc-1506 Santa Cruz

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
