# Peer review of "Expression of Adenosine Receptors in Rodent Pancreas"

_ijms, 2019, doi:10.3390/ijms20215329_

Round 1
Reviewer 1 Report
Overall, this study presents an interesting aim to explore the adenosine receptor expression in the exocrine pancreas. I present my comments/issues/criticisms below.
The introduction states on line 49, “In contrast to endocrine cells, few 49 studies have examined the function of adenosine receptors in the exocrine pancreas.” Why did this study only focus on the A2A and A2B receptors? I understand previous mRNA studies have suggested lower pancreatic levels of the A1 and A3, however, if the aim of the study (line 51) is to “ identify functional adenosine receptors involved in exocrine secretion of the pancreas.” then a more complete study would investigate all adenosine receptor subtypes to address this aim. Many of the studies references in the introduction refer to rodent experiments. What is known about adenosine receptor expression in primary human pancreatic tissue and how does this compare? This is especially important given that this study focuses entirely on rodents Further to the above comment, what is the rationale for looking at expression and function in different rodent species? The authors investigated islets (rat, mice), ducts (mice, guinea pigs) and secretion (rats). Has the author accurately validated the antibodies used for IHC? For example, through a western blot? Can they provide positive and negative controls for this study? The secretion data (figure 5), is hard to interpret in the absence of control curves. The authors state, line 143, that “The vehicle control (DMSO) did not influence in exocrine secretion (n = 3 rats; not shown).” This data is important to include for readers’ ability to compare between treatments. How were the secretion rates measured and statistical analyses on this data performed? This is not well explained The discussion reads more like an introduction, rather than a discussion of the data obtained and its relevance to the literature. I understand that the presence of adenosine receptors in endocrine/exocrine systems has important implications for numerous diseases, but it is not framed in the context of the current study, and thus is confusing for the reader. Moreover, the authors discuss dimers and homomers, where they say on line 204, “In the present study, exocrine secretion was stimulated significantly more by CGS 21680 than BAY 60-6583, indicating that homomeric A2A adenosine receptors were dominant in rat duct cells (Figure 5).” The data in this study cannot ascribe whether homomeric or dimeric receptors are responsible for any of the results obtained.Author Response
I am grateful to reviewer #1 for the careful comments that have helped me to improve the quality of the revised manuscript. As indicated in the responses that follow, I have taken all these comments into consideration in the revised version of the manuscript.
Comments from reviewer #1:
Comment 1:
The introduction states on line 49, “In contrast to endocrine cells, few studies have examined the function of adenosine receptors in the exocrine pancreas.” Why did this study only focus on the A2A and A2B receptors? I understand previous mRNA studies have suggested lower pancreatic levels of the A1 and A3, however, if the aim of the study (line 51) is to “identify functional adenosine receptors involved in exocrine secretion of the pancreas.” then a more complete study would investigate all adenosine receptor subtypes to address this aim.
Response 1:
I revised the Introduction on page 2, lines 53-55. “A2A and A2B adenosine receptors generally increase whereas A1 and A3 receptors decrease cAMP levels [18]. Thus, the present study focused on functional A2A and A2B adenosine receptors involved in ductal secretion of the pancreas.” I added additional reference: [18] Fredholm, B.B.; Irenius, E.; Kull, B.; Schulte, G. Comparison of the potency of adenosine as an agonist at human adenosine receptors expressed in Chinese hamster ovary cells. Biochem.Pharmacol.2001,61,443–448.
Comment 2:
Many of the studies references in the introduction refer to rodent experiments. What is known about adenosine receptor expression in primary human pancreatic tissue and how does this compare? This is especially important given that this study focuses entirely on rodents Further to the above comment, what is the rationale for looking at expression and function in different rodent species?
Response 2:
I revised the Introduction on page 2, lines 50-52. “Protein expression of A1 adenosine receptors was identified α-cells from the human pancreas [17]. However, the expression and function of adenosine receptors in primary human pancreatic duct cells are unknown.”
Comment 3:
The authors investigated islets (rat, mice), ducts (mice, guinea pigs) and secretion (rats). Has the author accurately validated the antibodies used for IHC? For example, through a western blot? Can they provide positive and negative controls for this study?
Response 3:
In order to verify the quality of the antibodies for A2Aand A2B adenosine receptors, I performed western blot analysis and now show the specific bands from the lysates of the isolated ducts in the new Fig. 5. For the negative control, I performed immunostaining using the isotype IgG and now show the negative staining in Fig. 1–4, I–L. Please see changes in the revised manuscript on page 2, lines 67-68; pages 2-3, lines 75-77; page 3, line 86; page 3, lines 94-96; page 4, lines 107-108; page 4, lines 118-120; page 4, lines 128-129; page 5, lines 138-140; page 5, lines 142-153; pages 11-12, lines 333-335; page 12, lines 338-356.
Comment 4:
The secretion data (figure 5), is hard to interpret in the absence of control curves. The authors state, line 143, that “The vehicle control (DMSO) did not influence in exocrine secretion (n = 3 rats; not shown).” This data is important to include for readers’ ability to compare between treatments.
Response 4:
I added data for the vehicle injected controls (DMSO) to new Fig. 6G–I. Please see changes in the revised manuscript on page 6, lines 173-174; page 7, lines 183-185.
Comment 5:
How were the secretion rates measured and statistical analyses on this data performed? This is not well explained.
Response 5:
I added the following sentence on page 7, lines 179-180. “Values were compared with the control value at 20 min. Pancreatic juice was collected in a silicone tube. Sample volumes were determined by the length of pancreatic juice in the silicone tube.” I revised the sentences on page 6, lines 157-163; page 12, lines 366-367.
Comment 6:
The discussion reads more like an introduction, rather than a discussion of the data obtained and its relevance to the literature.
Response 6:
I revised the Discussion on page 10, lines 240-244. “In pancreatic ducts, adenosine is produced by the hydrolysis of ATP that is secreted from acini by CCK stimulation [6]. Adenosine binds to adenosine receptors on the luminal membrane of duct cells and stimulates ductal secretion [22]. In the present study, A2A and A2B adenosine receptor antagonists showed a tendency to suppress 25% of the secretory rate by CCK stimulation (Figure 7), indicating acini-to-duct signaling.”
I added limitations in the Discussion on page 10, lines 268-270. “Several limitations of this study should be acknowledged. The protein level of A2A and A2B adenosine receptors could not be quantified by immunohistochemical analysis. The effects of CGS 21680 and BAY 60-6583 were small compared to those of secretin in in vivo experiments (Figure 6).”
Comment 7:
I understand that the presence of adenosine receptors in endocrine/exocrine systems has important implications for numerous diseases, but it is not framed in the context of the current study, and thus is confusing for the reader.
Response 7:
I deleted the sentences in new section on page 10 “3.2. Role of A2A and A2B Adenosine Receptors in Pancreas”. I revised the following sentences on page 10, line 266. “Future studies are required to clarify the intracellular signaling in pancreatic β cells.”
Comment 8:
Moreover, the authors discuss dimers and homomers, where they say on line 204, “In the present study, exocrine secretion was stimulated significantly more by CGS 21680 than BAY 60-6583, indicating that homomeric A2A adenosine receptors were dominant in rat duct cells (Figure 5).” The data in this study cannot ascribe whether homomeric or dimeric receptors are responsible for any of the results obtained.
Response 8:
I revised the following sentences on page 10, lines 271-281. “In the present study, ductal secretion was stimulated significantly more by CGS 21680 than BAY 60-6583 at the same concentration, indicating that A2A adenosine receptors were dominant in rat duct cells, whereas BAY 60-6583 showed a tendency to promote HCO3−-rich fluid secretion (Figure 6E,F). Thus, we cannot rule out the existence of A2B adenosine receptors in pancreatic duct cells. Previous studies postulated that A2Aand A2B adenosine receptors form functional heterooligomers [36]. The affinities of the A2A adenosine receptors for adenosine or CGS 21680 were shown to be decreased on co-expression with A2B adenosine receptors in recombinant cells [37]. While the affinity of A2B adenosine receptors for BAY 60-6583 was not altered by co-expression with A2A adenosine receptors. Future studies are needed in order to establish the presence of the heterooligomer in pancreatic duct cells and its functional relevance.”
I would like to thank the reviewer for his/her positive and helpful comments, and I hope that the manuscript is now considered acceptable for publication.
Reviewer 2 Report
In this work the authors first determined the expression of A2A receptor and A2B receptor in pancreatic tissues of rat and mouse. Then the effects of CGS-21680 and Bay60-6583 have been tested on exocrine secretion by measuring the secretory rate, concentration of protein and HCO3-. The authors conclude that these results showed that adenosine regulated exocrine secretion via A2A receptor in pancreatic duct cells.
However the work presents many important limitations and the results raise many questions. The experiments on A2A receptor expression and CGS are quite preliminary to fully sustain a role for A2A receptor on exocrine secretion. Experiments on dose-effects of CGS and with a selective antagonist of A2A receptor are missing and would strongly sustain a role for A2A in pancreatic secretion.
In Figure 1 the immunofluorescence signal of A2AR is quite diffuse in panel A while arrowhead, that indicate A2A in the cells membrane, suggest that its expression is very low. Isotype IgG control experiments are lacking in all immunofluorescence experiments.
In Figure 5 the authors show the results obtained with CGS and Bay on pancreatic secretion in rats. Here control experiments are not clearly described and must be shown. As indicated by arrows Sec and CCK are injected into rats after CGS injection or Bay injection. Relative vehicle injected controls should be shown. Moreover appropriate reference for secretin and CCk should be reported. In panel B and E the authors report the protein concentration. As stated in the paper CGS significantly reduced the protein concentration whilst Bay did not. However it seems that the delta decrease induced by CGS ( from 77.7 ± 8.4 to 41.2 ± 5.5 g/L) or Bay (102.9 ± 14.8 to 65.5 ± 5.9 g/L) is similar in these two set of experiments and the basal levels of protein concentrations are different. How do the authors explain these results?
Author Response
I am grateful to reviewer #2 for the careful comments that have helped me to improve the quality of the revised manuscript. As indicated in the responses that follow, I have taken all these comments into consideration in the revised version of the manuscript.
Comments from reviewer #2:
Comment 1:
However the work presents many important limitations and the results raise many questions. The experiments on A2A receptor expression and CGS are quite preliminary to fully sustain a role for A2A receptor on exocrine secretion. Experiments on dose-effects of CGS and with a selective antagonist of A2A receptor are missing and would strongly sustain a role for A2A in pancreatic secretion.
Response 1:
I performed additional experiments and added data for the application of selective antagonists of the adenosine receptors to new Fig. 7. I added new section: “2.7. Effect of Adenosine Receptor Antagonists on Pancreatic Secretion in Rats” in the Results on pages 7-8, lines 188-211. The A2A and A2B adenosine receptor antagonists showed a tendency to suppress 25% of the secretory rate by CCK stimulation, indicating acini-to-duct signaling. I added the following sentences on page 10, lines 268-271. “Several limitations of this study should be acknowledged. The protein level of A2A and A2B adenosine receptors could not be quantified by immunohistochemical analysis. The effects of CGS 21680 and BAY 60-6583 were small compared to those of secretin in in vivo experiments (Figure 6). Only the maximum dose of the chemical was tested.”
I revised the following sentence in the Conclusion on page 11, lines 304-305. “The results showed that A2A adenosine receptors may be, at least in part, involved in exocrine secretion of pancreatic duct cells via acini-to-duct signaling.”
I revised the following sentence in the Abstract on page 1, lines 17-19. “These results indicate that A2A adenosine receptors may be, at least in part, involved in exocrine secretion of pancreatic duct cells via acini-to-duct signaling.”
Comment 2:
In Figure 1 the immunofluorescence signal of A2AR is quite diffuse in panel A while arrowhead, that indicate A2A in the cells membrane, suggest that its expression is very low. Isotype IgG control experiments are lacking in all immunofluorescence experiments.
Response 2:
In order to verify the quality of the antibodies for A2Aand A2B adenosine receptors, I performed immunostaining using the isotype IgG and now show the negative staining in Fig. 1–4, I–L. Please see changes in the revised manuscript on page 2, lines 67-68; pages 2-3, lines 75-77; page 3, line 86; page 3, lines 94-96; page 4, lines 107-108; page 4, lines 118-120; page 4, lines 128-129; page 5, lines 138-140; pages 11-12, lines 333-335.
Comment 3:
Here control experiments are not clearly described and must be shown. As indicated by arrows Sec and CCK are injected into rats after CGS injection or Bay injection. Relative vehicle injected controls should be shown.
Response 3:
I added data for the vehicle injected controls (DMSO) to new Fig. 6G–I. Please see changes in the revised manuscript on page 6, lines 173-174; page 7, lines 183-185.
Comment 4:
Moreover appropriate reference for secretin and CCk should be reported.
Response 4:
I added additional reference on page 6, lines 173: [21] Dockray, G.J. The action of secretin, cholecystokinin-pancreozymin and caerulein on pancreatic secretion in the rat. J. Physiol.1972, 225, 679–692.
Comment 5:
In panel B and E the authors report the protein concentration. As stated in the paper CGS significantly reduced the protein concentration whilst Bay did not. However it seems that the delta decrease induced by CGS ( from 77.7 ± 8.4 to 41.2 ± 5.5 g/L) or Bay (102.9 ± 14.8 to 65.5 ± 5.9 g/L) is similar in these two set of experiments and the basal levels of protein concentrations are different. How do the authors explain these results?
Response 5:
I revised the following sentence on page 6, lines 166-167. “However, the protein concentration showed a tendency to decrease from 102.9 ± 14.8 to 65.5 ± 5.9 g/L (Figure 6E; P= 0.07), indicating ductal secretion.”
I would like to thank the reviewer for his/her positive and helpful comments, and I hope that the manuscript is now considered acceptable for publication.
Reviewer 3 Report
The current manuscript report important data on adenosine expression in the pancreas. Why the authors haven't presented the A3 adenosine receptor and the A1?
Also in the conclusion, the authors state that data show a potential utilization of A2A as an anti-cancer target, however it is definitely not based on any data presented in the current manuscript.
Author Response
I am grateful to reviewer #3 for the careful comments that have helped me to improve the quality of the revised manuscript. As indicated in the responses that follow, I have taken all these comments into consideration in the revised version of the manuscript.
Comments from reviewer #3:
Comment 1:
The current manuscript report important data on adenosine expression in the pancreas. Why the authors haven't presented the A3 adenosine receptor and the A1?
Response 1:
I revised the Introduction on page 2, lines 53-55. “A2A and A2B adenosine receptors generally increase whereas A1 and A3 receptors decrease cAMP levels [18]. Thus, the present study focused on functional A2A and A2B adenosine receptors involved in ductal secretion of the pancreas.” I added additional reference: [18] Fredholm, B.B.; Irenius, E.; Kull, B.; Schulte, G. Comparison of the potency of adenosine as an agonist at human adenosine receptors expressed in Chinese hamster ovary cells. Biochem.Pharmacol.2001,61,443–448.
Comment 2:
Also in the conclusion, the authors state that data show a potential utilization of A2A as an anti-cancer target, however it is definitely not based on any data presented in the current manuscript.
Response 2:
I performed additional in silico analysis and presented data, which ADORA2B (A2B adenosine receptor) expression was associated with poor disease-free survival of human pancreatic adenocarcinoma patients in new Fig. 8. Please see changes in the revised manuscript on page 1, lines 19-20; page 9, lines 212-231; pages 10-11, lines 286-289; page 11, lines 305-306; page 13, lines 373-380.
I added additional references on page 13, line 375: [46,47]
[46] Cerami, E.; Gao, J.; Dogrusoz, U.; Gross, B.E.; Sumer, S.O.; Aksoy, B.A.; Jacobsen, A.; Byrne, C.J.; Heuer, M.L.; Larsson, E.; et al.The cBio cancer genomics portal: an open platform for exploring multidimensional cancer genomics data. Cancer Discov.2012, 2, 401–404.
[47] Gao, J.; Aksoy, B.A.; Dogrusoz, U.; Dresdner, G.; Gross, B.; Sumer, S.O.; Sun, Y.; Jacobsen, A.; Sinha, R.; Larsson, E.; et al.Integrative analysis of complex cancer genomics and clinical profiles using the cBioPortal. Sci Signal.2013,6, pl1.
I would like to thank the reviewer for his/her positive and helpful comments, and I hope that the manuscript is now considered acceptable for publication.
Round 2
Reviewer 1 Report
Manuscript has been sufficiently amended
Reviewer 2 Report
The revised version of the manuscript is acceptable.
Reviewer 3 Report
No comments. This now can be accepted.